# Nitrogen-Fixing *Paenibacillus haidiansis* and *Paenibacillus sanfengchensis*: Two Novel Species from Plant Rhizospheres

**DOI:** 10.3390/microorganisms12122561

**Published:** 2024-12-12

**Authors:** Weilong Zhang, Miao Gao, Rui Hu, Yimin Shang, Minzhi Liu, Peichun Lan, Shuo Jiao, Gehong Wei, Sanfeng Chen

**Affiliations:** 1College of Biological Sciences, China Agricultural University, Beijing 100193, China; zwlong2022@163.com (W.Z.); hurui_19991298@163.com (R.H.); drasym@163.com (Y.S.); lmz1453145198@163.com (M.L.); peichun_lan@163.com (P.L.); 2State Key Laboratory of Efficient Utilization of Arid and Semi-Arid Arable Land in Northern China, Key Laboratory of Microbial Resources Collection and Preservation, Institute of Agricultural Resources and Regional Planning, Chinese Academy of Agricultural Sciences, Beijing 100081, China; gaomiao@caas.cn; 3State Key Laboratory for Crop Stress Resistance and High-Efficiency Production, College of Life Sciences, Northwest A&F University, Yangling 712100, China; shuojiao@nwafu.edu.cn (S.J.); weigehong@nwafu.edu.cn (G.W.)

**Keywords:** *Paenibacillus* sp. nov., nitrogen-fixing bacteria, genomic characteristics, phylogenetic analysis

## Abstract

Two strains, M1 and H32 with nitrogen-fixing ability, were isolated from the rhizospheres of different plants. Genome sequence analysis showed that a *nif* (*ni*trogen *f*ixation) gene cluster composed of nine genes (*nifB nifH nifD nifK nifE nifN nifX hesA nifV*) was conserved in the two strains. Phylogenetic analysis based on the 16S rRNA gene sequence showed that strains M1 and H32 are members of the genus *Paenibacillus*. Strains M1 and H32 had 97% similarity in the 16S rRNA gene sequences. Strain M1 had the highest similarity (97.25%) with *Paenibacillus vini* LAM 0504T in the 16S rRNA gene sequences. Strain H32 had the highest similarity (97.48%) with *Paenibacillus faecis* TCIP 101062T in the 16S rRNA gene sequences. The average nucleotide identity (ANI) and digital DNA–DNA hybridization (dDDH) values between strain M1 and its closest member *P. vini* were 78.17% and 22.3%, respectively. ANI and dDDH values between strain H32 and its closest member *P. faecis* were 88.94% and 66.02%, respectively. The predominant fatty acid of both strains is anteiso-C15:0. The major polar lipids of both strains are DPG (diphosphatidylglycerol) and PG (phosphatidylglycerol). The predominant isoprenoid quinone of both strains is MK-7. With all the phylogenetic and phenotypic divergency, two novel species *Paenibacillus haidiansis* sp. nov and *Paenibacillus sanfengchensis* sp. nov are proposed with the type strain M1^T^ [=GDMCC (Guangdong Culture Collection Centre of Microbiology) 1.4871 = JCM (Japan Collection of Microorganisms) 37487] and with type strain H32^T^ (=GDMCC 1.4872 = JCM37488).

## 1. Introduction

*Paenibacillus* is a large genus of Gram-positive and endospore-forming bacteria. The genus *Paenibacillus*, which was originally included within the genus *Bacillus*, was reclassified as a separate genus in 1993 [1]. *Paenibacillus* (formerly *Bacillus*) *polymyxa* was proposed as the type species. At that time, the genus *Paenibacillus* encompassed 11 species, including the three N_2_-fixing species *Paenibacillus polymyxa*, *Paenibacillus azotofixans* and *Paenibacillus macerans.* The genus *Paenibacillus* currently comprises more than 400 named species (https://lpsn.dsmz.de/genus/paenibacillus, 1 September 2024). Most of the members of the genus *Paenibacillus* are non-nitrogen-fixing bacteria and only a few dozen species are able to fix nitrogen. For example, the nitrogen-fixing *Paenibacillus* species include *Paenibacillus sinensis*, *Paenibacillus caui*, *Paenibacillus sonchi*, *Paenibacillus jilunlii*, *Paenibacillus sophorae*, *Paenibacillus zanthoxyli*, *Paenibacillus sabinae*, *Paenibacillus forsythiae*, *Paenibacillus triticisoli* and *Paenibacillus taohuashanense* [2,3,4,5,6,7,8,9,10,11].

*Paenibacillus* is an important source of plant growth-promoting rhizosphere bacteria (PGPR), which can directly promote plant growth via nitrogen fixation, phosphate solubilization, phytohormone production and secretion of siderophores [12]. Some species of *Paenibacillus* produce antimicrobial compounds, such as Fusaricidins and Polymyxin, that can inhibit plant pathogenic fungi and also induce plant disease resistance [13,14,15]. Nitrogen-fixing *Paenibacillus triticisoli*, used as biofertilizer in wheat and maize, enhanced grain yields and changed the structure of microbial populations [16,17]. Comparative genome-sequencing analysis revealed that a *nif* (*ni*trogen *f*ixation) gene cluster composed of 9–10 genes [*nifB nifH nifD nifK nifE nifN nifX hesA* (*orf1*) *nifV*] encoding Mo-nitrogenase is conserved in N_2_-fixing *Paenibacillus* strains [18].

Nitrogen is an essential nutrient element for plants. Cultivation of cereals (e.g., rice, wheat, maize), vegetables and fruit trees depend highly on chemical nitrogen fertilizer. However, overuse of chemical nitrogen fertilizer causes grave environmental problems. Biological nitrogen fixation is an alternative to the use of chemical nitrogen fertilizers. Nitrogen fixation performed by microorganisms in the rhizosphere is a significant source of nitrogen in terrestrial systems. Thus, we aim to isolate nitrogen-fixing *Paenibacillus* strains from the plant rhizosphere, so that the nitrogen-fixing bacteria can be used as bio-fertilizer in agriculture.

In this study, two nitrogen-fixing strains, M1 and H32, were isolated from the rhizospheres of *Hosta plantaginea* and *Oryza sativa* (rice), respectively. Nitrogenase activity and *nifH* gene sequencing analysis demonstrated that the two strains are N_2_-fixing bacteria. Both strains M1 and H32 are proposed as the two novel species of *Paenibacillus* genus, based on the data of 16S rDNA phylogenetic analysis, average nucleotide identity (ANI), digital DNA–DNA hybridization (dDDH), phenotypic characteristics and compositions of the cell fatty acids, the major polar lipids and the predomin isoprenoid quinone.

## 2. Materials and Methods

### 2.1. Isolation of Strains

Strain M1 was isolated from the rhizosphere soil of *Hosta plantaginea* in Haidian District of Beijing, China (39°57′52.84″ N 116°17′52.84″ E). The soil type is sandy loam. The pH is 6.0–7.5. The organic matter content is about 15 g/kg. Strain H32 was isolated from the rhizosphere soil of *Oryza sativa* L. ssp. *Japonica cultivar Nipponbare* (rice) in Hebei province of China (N 39°28′42″–32′, E 116°38′07″–44′06″). The pH of soil in paddy fields is 6.5–7.5. The organic matter content is about 29 g/kg. For isolation of N_2_-fixing strains, soil sample was incubated on nitrogen-free medium, containing (per liter) 20 g sucrose, 0.1 g K_2_HPO_4_, 0.4 g KH_2_PO_4_, 0.2 g MgSO_4_·7H_2_O, 0.1 g NaCl, 0.01 g FeCl_3_ and 0.002 g Na_2_MoO_4_. After 3 days incubation at 30 °C, strains M1 and H32 were screened by subculturing onto the same medium.

### 2.2. Nitrogenase Activity Assay

For nitrogenase activity assay, bacterial strains were grown in nitrogen-free medium. The nitrogenase activity was determined using the acetylene reduction assay and expressed as nmol C_2_H_2_/mg protein/h as described previously [4]. *Paenibacillus polymyxa* WLY78 was used as a positive control for nitrogenase activity assay [18].

### 2.3. nifH Gene Sequence and Phylogenetic Analysis

The *nifH* coding region of strains M1 and H32 was PCR amplified with primers *nifH* P1 (5′-GGCTGCGATCCVAAGGCCGAYTCVACCCG-3′) and *nifH* P2 (5′-CTGVGCCTTGTTYTCGCGGATSGGCATGGC-3′). The amplified *nifH* products were sequenced by SinoGenoMax in China. The sequencing results were then subjected to homology analysis using BLASTN on NCBI (https://blast.ncbi.nlm.nih.gov, 1 September 2024), and a phylogenetic tree was constructed by using the maximum likelihood (ML) in the software MEGA 7 [19].

### 2.4. 16S rRNA Gene Sequence and Phylogenetic Analysis

16S rRNA gene sequences of both strains M1 an H32 were amplified using forward primer 16S P1 (5′-AGAGTTTGATCCTGGCTCAGAACGAACGCT-3′) and reverse primer 16S P2 (5′-TACGGCTACCTTGTTACGACTTCACCCC-3′) [20]. The 16S rRNA gene products were sequenced by SinoGenoMax in Beijing, China. The closest phylogenetic species were derived from the 16S rRNA gene sequences in the NCBI server [21]. Generation of the phylogenetic tree was performed using the maximum likelihood (ML) in the software MEGA 7 [22].

### 2.5. Genome Sequencing and Analysis

Genomic DNA was extracted using the TIANamp Bacteria DNA Kit (DP302-02) provided by TIANGEN BIOTECH Co., Ltd., Beijing, China. Genomes of strains M1 and H32 were sequenced in Novogene Co., Beijing, China by using the Illumina PE150 platform. Fragments below 500 bp were filtered out, and contaminated samples were further decontaminated. The assembled genome was then evaluated, statistically analyzed, and subject to subsequent gene prediction. The GeneMarkS software (Version 4.17) was employed to predict protein-coding genes in the newly sequenced genome [23]. The protein sequences of predicted genes were aligned with various functional databases using Diamond (e value ≤ 1 × 10^−5^). The whole-genome similarity was assessed using the Average Nucleotide Identity (ANI) tool with the OrthoANIu algorithm [https://www.ezbiocloud.net/tools/ani, 1 September 2024]. Digital DNA–DNA hybridization (dDDH) values were computed at GGDC (Genome-to-Genome Distance Calculator) [24].

### 2.6. Morphological, Physiological and Biochemical Analysis

Bacterial strains were routinely grown on LB agar at 30 °C for 2–3 days. Transmission Electron Microscopy (TEM) was used for morphological observation. Gram staining was performed using the Color Gram 2 kit (bioMérieux, Tokyo, Japan). Physiological and biochemical analyses, such as NaCl tolerance and growth at different pH values, were performed as described by Li et al. [2].

### 2.7. Chemotaxonomic Characterization

The compositions of whole-cell fatty acids, polar lipids and respiratory quinones were analyzed at the Institute of Agricultural Resources and Regional Planning, Chinese Academy of Agricultural Sciences, Beijing, China. Polar lipid was extracted according to the method described by Minnikin et al. [25] and was identified by two-dimensional TLC (Thin Layer Chromatography). The compositions of cellular fatty acids were analyzed using the method described by Komagata and Suzuki [26] using Sherlock Identification System (MIDI) [27]. Cellular menaquinones and isoprenoid quinones were extracted and analyzed by HPLC (High Performance Liquid Chromatography) [28].

## 3. Results

### 3.1. Strain Isolation, Nitrogenase Activity and nifH Genes

Two strains M1 and H32 were isolated by screening on nitrogen-free medium from the rhizospheres of plants grown in different regions of China. Strain M1 was isolated from the rhizosphere of *Hosta plantaginea* in Beijing suburb, China, and strain H32 was isolated from the rhizosphere of rice in Hebei province, China.

To confirm the nitrogen-fixing ability of the two strains M1 and H32, nitrogenase activity and *nifH* gene were determined. Nitrogenase activity was determined by using acetylene-reduction assay. Strains M1 and H32 exhibited nitrogenase activity with 2950 (nmol C_2_H_2_/mg protein/h) and with 500 (nmol C_2_H_2_/mg protein/h), respectively. *P. polymyxa* WLY78 (positive control) had nitrogenase activity with 2100 (nmol C_2_H_2_/mg protein/h). PCR analysis showed that the two strains M1 and H32 had *nifH* genes. Phylogenetic analysis based on *nifH* gene sequences revealed that the two strains M1 and H32 clustered with the *nifH* genes of *Paenibacillus* strains (Figure 1).

### 3.2. Phylogenetic Analysis of 16S rRNA Gene

A nearly full-length 16S rRNA gene (ca. 1500 bp) was PCR amplified from each of the two strains M1 and H32 and then was sequenced. A comparison of the 16S rRNA gene sequences of the two strains M1 and H32 with the sequences held in the GenBank database revealed that the two strains belong to the genus *Paenibacillus* (Figure 2). Strain M1 showed a 97% identity with H32 at 16S rRNA gene sequences. Strain M1 had the highest (97.25%) identity with *P. vini* LAM 0504T and had a lower than 97% identity with other *Paenibacillus* species (such as *Paenibacillus timonensis*, *Paenibacillus motobuensis*, *Paenibacillus azoreducens*) at 16S rRNA gene sequences. Strain H32 had the highest (97.48%) identity with *Paenibacillus faecis* TCIP 101062T and had a lower than 97% identity with other *Paenibacillus* species (such as *Paenibacillus physcomitrellae*, *Paenibacillus yonginensis*, *Paenibacillus konsidensis*) at 16S rRNA gene sequences. Generally, 98.65% similarity of 16S rRNA gene sequences is suggested as the threshold to differentiate bacterial species [29]. The current results that the 16S rRNA gene sequence similarity of strains M1 and H32 with their closest species was below the threshold value of 98.65% indicate that both strains M1 and H32 are two novel species within the genus *Paenibacillus*.

### 3.3. Genome Sequence and Similarity Analysis

The two strains M1 and H32 were genome-sequenced to evaluate their relatedness to the closely related recognized species in the genus *Paenibacillus*. The genome size of strain M1 was 5.7 Mb which has 5426 genes. There are 87 tRNA genes and two 16S rRNA genes in the M1 strain. The DNA G+C content of strain M1 is 52% (Table 1). Whereas its closely related strain *P. vini* had a complete genome size of 5.6 Mb which has 5274 genes. The DNA G+C content of *P. vini* is 49%.

The genome size of strain H32 was 6.48 Mb carrying 5511 genes. The G+C content of strain H32 was 52.99% (Table 1). Whereas its closely related strain *P. faecis* had a complete genome size of 6.3 Mb comprising 5699 genes. The DNA G+C content of *P*. *faecis* is 53%. The genome sequences of the two strains M1 and H32 were deposited in GenBank under accession numbers GCA_036553665.1 and GCA_036864765.1, respectively.

The average nucleotide identity (ANI) and digital DNA-DNA hybridization (dDDH) values are generally used to define bacterial species [30]. The ANI for the species threshold is 95% and DDH for the species threshold is 70% [30,31]. The ANI and dDDH values between strain M1 and the reference strain *P. vini* are 78.17% and 22.3%, respectively. The ANI and dDDH values between strain H32 and the reference strain *P. feacis* DSM 23593 were 88.94% and 66.20%, respectively. These genomic relatedness data are below the thresholds of ANI (95.0%) and dDDH (70.0%), indicating that strains M1 and H32 are genomically distinct from their closest species of the genus *Paenibacillus*.

### 3.4. Analysis of Nitrogen Fixation and Nitrogen Metabolism Genes

Genome sequence analysis demonstrated that each genome of the two strains M1 and H32 contains a compact *nif* gene cluster composed of nine genes (*nifB*, *nifH*, *nifD*, *nifK*, *nifE*, *nifN*, *nifX*, *hesA*, *nifV*) encoding Mo-nitrogenase which is conserved in N_2_-fixing *Paenibacillus* strains [18]. The *nif* gene cluster carried in the genomes of the two strains M1 and H32 is consistent with the nitrogenase activity observed in both stains.

In addition to the *nif* genes, each genome of the two strains M1 and H32 contains two sets of *nasE*, *nasD* and *nasB* encoding nitrite reductase. There is a *nirK* encoding nitrite reductase, a *nirC* encoding nitrite transporter and a *nasA* encoding nitrate/nitrite transporter in both strains. There are two copies of *glnA* encoding glutamine synthetase, one of which is linked together with *glnR* as an operon *glnRA*, the other one of which is separated on the genome site [32]. GlnR encoded by *glnR* is a global transcription regulator controlling the expression of *nif* genes and some other genes involved in nitrogen metabolism in *P. ploymyxa* [32,33,34]. There is a *gdhA* encoding glutamate dehydrogenase in both strains M1 and H32, and there are *gltD* and *gltB* encoding glutamate synthase (NADPH) small chain and large chains, respectively. There is a *gltC* encoding transcription activator of glutamate synthase operon in both strains.

### 3.5. Phenotypic Characteristics

The two strains M1 and H32 are Gram-positive, facultatively anaerobic, motile and rod-shaped. Both strains have peritrichous flagella on the cell surface observed under a transmission electron microscope (Figure 3).

Strains M1 and H32, together with their reference strains *P. vini* LAM0504T and *P. faecis* 65684T, were tested for a range of physiological and biochemical characteristics. As shown in Table 2, Both strain M1 and the reference strain *P. vini* LAM0504T exhibited positive nitrate reductase activity. Both strain M1 and the reference strain *P. vini* LAM0504T utilized xylose, glucose, trehalose, galactose and maltose to produce acid. However, both strain M1 and the reference strain *P. vini* LAM0504T showed variation in fructose utilization and NaCl tolerance.

Both strain H32 and the reference strain *P. faecis* 65684T had nitrate reductase activity and utilized fructose, xylose, glucose, trehalose, galactose and maltose to produce acid. However, both strain H32 and the reference *P. faecis* 65684T showed variation in the utilization of D-mannitol and L-rhamnose and pH range tolerance.

### 3.6. Chemotaxonomic Analyses

Chemotaxonomic analyses included the cellular fatty acid, polar lipids and isoprenoid quinone that are used in the classification of bacteria [29,30,31,32]. As shown in Table 3, anteiso-C15:0 was the major fatty acid component of strains M1 (45.71%) and H32 (61.41%), and anteiso-C15:0 was also the predominant fatty acid of their reference strains *P. vini* LAM0504T (50.30%) and *P. faecis* DSM 23593T (45.42%). Our results are consistent with the findings that anteiso-C15:0 is the predominant fatty acid of the genus *Paenibacillus* [2,3,4,5,6,7,8,9,10,11]. Although anteiso-C15:0 is the major fatty acid for both strains (M1 and H32) and their reference strains, the fatty acid contents show significant variation among these strains (Table 3). These data indicate that strains M1 and H32 are different from each other and are distinguished from other members of the genus *Paenibacillus*.

The major polar lipids were detected by two-dimensional TLC. The major polar lipids of strain M1 are DPG (diphosphatidylglycerol), PG (phosphatidylglycerol), AL (aminolipid) and two unidentified phospholipids (PL3 and PL4). Whereas, the major polar lipids of strain H32 are DPG, PG, PE (phosphatidylethanolamine) and APL (aminophospholipids) (Appendix A). The polar lipid profiles of both strains M1 and H32 are similar to those of other species of the genus *Paenibacillus* [11]. Respiratory quinone is mainly composed of ubiquinone (Q) and menaquinone (MK). The major respiratory quinone component of both strains M1 and H32 is menaquinone-7 (MK-7) (Appendix A), in conformity with the genus *Paenibacillus*.

### 3.7. Description of Paenibacillus haidiansis sp. nov

*Paenibacillus haidiansis* (haidian.sis. L.gen. n. Hardian of Beijing, where the type strain M1 was isolated). Type strain M1 is Gram-positive, facultative anaerobic and rod-shaped (0.3–0.5 μm × 2.5–3.5 μm). Cells have peritrichous flagella. Ellipsoidal spores are located centrally in swollen sporangia. The colonies on LB medium are white, convex, circular and translucent. Cells grow at 25–35 °C (optimum, 30 °C), pH 6.0–8.0 (optimum, 7.0), and 0–1.0% NaCl concentrations. Nitrate is reduced to nitrite. Starch is hydrolyzed. The genome of strain M1 has nine genes (*nifB, nifH, nifD, nifK, nifE, nifN, nifX, hesA, nifV*) encoding Mo-nitrogenase. Strain M1 is able to reduce N_2_ to NH_3_ (i.e., biological nitrogen fixation). Able to utilize D-fructose, D-xylose, D-glucose, trehalose, D-galactose, maltose to produce acid, but not able to utilize L-rhamnose, D-mannitol, D-sorbitol and inositol. The predominant fatty acid is anteiso-C15:0. The major polar lipids are DPG, PG and AL. The only menaquinone is MK-7. The DNA G+C content of this strain is 52%.

The type strain, M1^T^ (=CGMCC 1.4871 = JCM 37487, was isolated from the rhizosphere soil of *Hosta plantantae* in the Haidian District of Beijing, China. The GenBank accession numbers for the 16S rRNA sequence is PP204203.1 and for the draft genome sequence is GCA_036553665.1.

### 3.8. Description of Paenibacillus sanfengchensis nov sp. nov

*Paenibacillus sanfengchensis* (sanfeng′chen.sis. L.gen. n. *sanfengchensis* of Sanfeng Chen, named after Professor Sanfeng Chen, China). Type strain H32 is Gram-positive, facultatively anaerobic, and rod-shaped (0.3.5–0.5 μm × 2.5–3.0 μm). Cells have peritrichous flagella. Ellipsoidal spores are located centrally in swollen sporangia. The colonies on LB medium are white, convex, circular and translucent. The genome of strain H32 has nine genes (*nifB, nifH, nifD, nifK, nifE, nifN, nifX, hesA, nifV*) encoding Mo-nitrogenase. Strain H32 is able to reduce N_2_ to NH_3_ (i.e., biological nitrogen fixation). Cells grow at 20–40 °C (optimum, 30 °C), pH 5.0–9.0 (optimum, 7.0), and 0–1.0% NaCl concentrations. Starch is hydrolyzed. Nitrate is reduced to nitrite. Able to utilize D-fructose, D-xylose, D-glucose, trehalose, D-galactose, maltose, L-rhamnose and D-mannitol to produce acid, but not able to utilize D-sorbitol and inositol. The predominant fatty acid is anteiso-C15:0. The major polar lipids are DPG, PG, PE and APL. The only menaquinone is MK-7. The DNA G+C content of this strain is 52.5%.

The type strain, H32^T^ (=CGMCC 1.4872 = JCM 37488, was isolated from the rhizosphere soil of *Oryza sativa* (rice) in Hebei province, China. The GenBank accession number for the 16S rRNA sequence is P204203.1 and for the draft genome sequence is GCA_036864765.1.

## 4. Discussion

*Paenibacillus* is a large genus of Gram-positive, facultative anaerobic and endospore-forming bacteria. N_2_-fixng *Paenibacillus* species and strains have potential uses as a bacterial fertilizer in agriculture.

In this study, two strains M1 and H32 were isolated from the rhizospheres of *Hosta plantaginea* and *Oryza sativa* (rice), respectively. The two strains are Gram-positive, facultative anaerobic and endospore-forming bacteria. Nitrogenase activity assay and *nifH* sequencing analysis showed that the two strains are N_2_-fixing bacteria. Genome sequence analysis revealed that each genome of the two strains has a *nif* gene cluster composed of nine genes (*nifB nifH nifD nifK nifE nifN nifX hesA nifV*) that are characteristics of N_2_-fixing *Paenibacillus* strains [18].

A comparison of the 16S rRNA gene sequences of the two strains M1 and H32 with the sequences held in the GenBank database revealed that the two strains belong to the genus *Paenibacillus.* Strain M1 showed 97% identity with strain H32 at 16S rRNA gene sequences. Strain M1 had the highest (97.25%) identity with *P. vini* and had a lower than 97% identity with other named *Paenibacillus* species at 16S rRNA gene sequences. Strain H32 had the highest (97.48%) identity with *P. faecis* and had a lower than 97% identity with other named *Paenibacillus* species. ANI and dDDH values between strain M1 and its closest member *P. vini* were 78.17% and 22.3%, respectively. The ANI and dDDH values between strain H32 and its closest member *P. faecis* were 88.94% and 66.02%, respectively. The predominant fatty acid is anteiso-C15:0 for both strain M1 (45.71%) and H32 (61.41%). The predominant isoprenoid quinone is MK-7 for both strains M1 and H32. The two strains M1 and H32 exhibited some variation with their reference strains in the contents and components of the cellular fatty acids and polar lipids. These data suggest that strains M1 and H32 are two novel species within the genus *Paenibacillus*. Thus, two novel species *Paenibacillus haidiansis* and *Paenibacillus sanfengchensis* are proposed.

It is known that most of the members of the genus *Paenibacillus* are non-nitrogen-fixing bacteria and only a few dozen species are able to fix nitrogen. In this study, two nitrogen-fixing novel species *Paenibacillus haidiansis* and *Paenibacillus sanfengchensis* are obtained, providing bacterial strains for study and utilization of biological nitrogen fixation. The genus *Paenibacillus*, which was originally included within the genus *Bacillus*, was reclassified as a separate genus in 1993 [1]. When the genus *Paenibacillus* was created, the three N_2_-fixing species *Bacillus polymyxa*, *Bacillus azotofixans* and *Bacillus macerans* were transferred to this new genus. Recently, we tried to isolate N_2_-fixing *Bacillus* strains but failed. The results support that the N_2_-fixing and endospore-forming bacteria mainly belong to the genus *Paenibacillus*. It is well known that most N_2_-fixing bacteria (e.g., *Azotobacter vinelandii, Klebsiella oxytoca*) are Gram-negative. In most of the Gram-negative and N_2_-fixing bacteria, the expression of *nif* genes is regulated by the transcription regulator NifA [35]. Whereas GlnR regulates expression of *nif* genes in Gram-positive and N_2_-fixing *Paenibacillus* strains. Gram-positive and N_2_-fixing *Paenibacillus* strains and the Gram-negative and N_2_-fixing bacteria also show some differences in colonization, nitrogen-fixation efficiency and growth rate.

It is now recognized that the free-living and plant-associated N_2_-fixing microorganisms play a contributing role in the growth of non-legumes by N_2_ fixation. Inoculation of non-legume plants with N_2_-fixing biofertilizers (also known as biological inoculants) could help to increase the available nitrogen content of the crop and reduce the usage amount of chemical nitrogen fertilizer in a variety of agricultural systems [36,37,38,39]. In the near future, we will utilize *P. haidiansis* M1 and *P. sanfengchensis* H32 to inoculate plants (e.g., rice, wheat, maize, cucumber, tomato) in a greenhouse, and then evaluate their effects on promoting plant growth through N_2_-fixation. If both strains or one of them have efficient effects on promoting plant growth through N_2_-fixation, both or one of them could be used as N_2_-fixing biofertilizers in agriculture on a large scale.

## 5. Conclusions

In this study, two strains M1 and H32 were isolated from the rhizospheres of *Hosta plantaginea* and *Oryza sativa* (rice), respectively. Both strains M1 and H32 are able to fix nitrogen. Based on the phenotypic, phylogenetic, genomic and chemotaxonomic dissimilarity, two novel species *Paenibacillus haidiansis* sp. nov and *Paenibacillus sanfengchensis* sp. nov are proposed with the type strain M1^T^ = CGMCC 1.4871 = JCM 37487 and with the type strain H32^T^ (=CGMCC 1.4872 = JCM 37488.

## Figures and Tables

**Figure 1 microorganisms-12-02561-f001:**
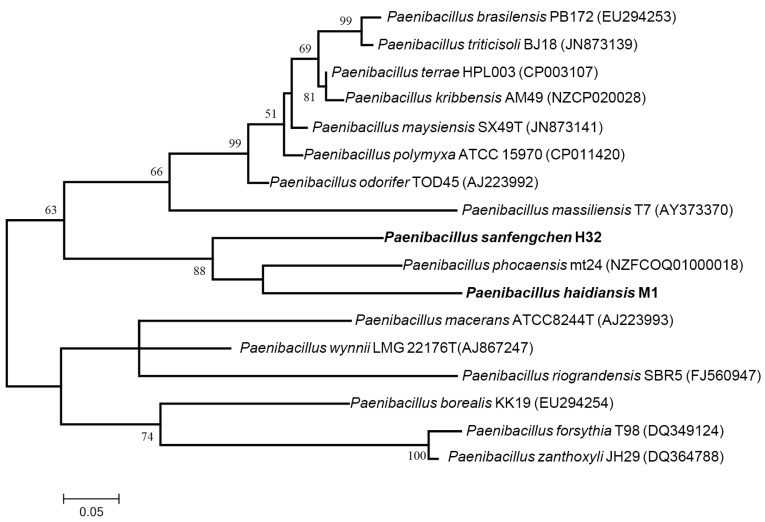
The maximum phylogenetic likelihood tree based on *nifH* gene sequences (300 bp) of the novel strains M1 and H32, together with their closely related taxonomic groups. Bootstrap analyses were performed with 1000 cycles. Numbers (50%) at nodes are bootstrap values. Bar 0.05 substitutions per nucleotide position.

**Figure 2 microorganisms-12-02561-f002:**
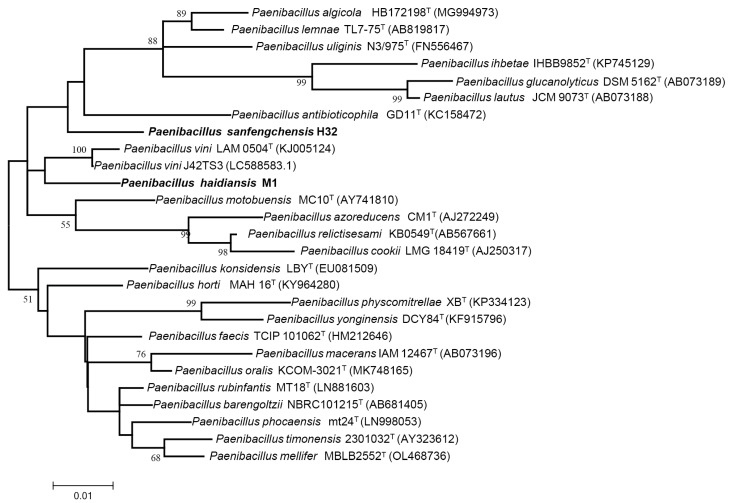
The maximum likelihood phylogenetic tree based on the 16S rRNA gene sequences of the novel strains M1 and H32, together with their closely related taxonomic groups. Bootstrap analyses were performed with 1000 cycles. Numbers (50%) at nodes are bootstrap values. Bar 0.05 substitutions per nucleotide position.

**Figure 3 microorganisms-12-02561-f003:**
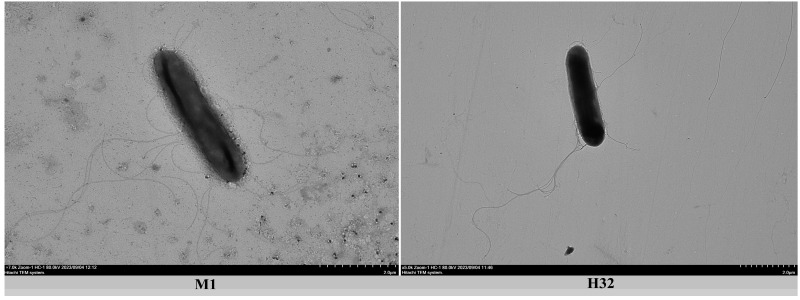
Transmission electron microscopic images of vegetative cell and flagella of strains M1 and strain H32.

**Table 1 microorganisms-12-02561-t001:** General genomic information of strains M1, H32 and their closely related type strains.

Strains	Accession No.	Genome Size (Mb)	DNA GC (%)	Gene Number	RNAs		
					tRNA	16S rRNAs	ncRNA (Non-Coding RNA)
Test strain M1	GCA_036553665.1	5.7	52	5462	87	2	4
Type strain *Paenibacillus vini* strain J42TS3	GCA_018403325.1	5.6	49	5274	93	2	4
Type strain *Paenibacillus vini* strain CENA-BCM001	GCA_030412165.1	5.7	49	5293	93	2	4
Test strain H32	GCA_036864765.1	6.46	52.5	5511	77	1	8
Type strain *Paenibacillus faecis* strain J25TS5	GCA_008084145.1	6.3	53	5743	83	2	4
Type strain *Paenibacillus faecis* DSM 23593(T)	GCA_008084145.1	6.3	53	5699	64	4	4

**Table 2 microorganisms-12-02561-t002:** Differential characteristic features of test strains M1 and H32 from their reference strains.

Characteristic	Test Strain	Reference Strain	Test Strain	Reference Strain
	M1	*P. vini* LAM0504^T^	H32	*P. faecis* 65684^T^
pH range	6–8	6–8	5–9	6–9
NaCl	0–1.0%	0–3%	0–1.0%	0–3%
Growth temperature (°C)	25–35	30	20–40	30
Nitrate reduction	+	+	+	+
Starch hydrolysis	+	+	+	+
Dihydroxyacetone production	−	−	−	−
Mobility	+++	+	+	+
Flagellum	+	+	+	+
Production of acid from following substrates:				
D-fructose	+	−	+	+
D-xylose	+	+	+	+
D-glucose	+	+	+	+
Trehalose	+	+	+	+
D-galactose	+	+	+	+
Maltose	+	+	+	+
D-mannitol	−	−	−	+
L-rhamnose	−	−	−	+
D-sorbitol	−	−	−	−
Inositol	−	−	−	−

+, positive; −, negative.

**Table 3 microorganisms-12-02561-t003:** The cellular fatty acid content (% of the totals) of strains M1 and H32, together with their closely related reference strains *P. vini* LAM0504T and *P. faecis* DSM 23593T.

Fatty Acid	M1	*P. vini* LAM0504^T^	H32	*P. faecis* DSM 23593^T^
Saturated				
C_12:0_	0.62	0.5	1.61	1.60
C_14:0_	3.16	3.30	1.10	1.77
C_16:0_	24.39	9.73	6.47	21.56
C_17:0_	TR	-	TR	1.50
Unsaturated				
C14:1 ω5c	-	0.17	TR	TR
C_19:0_ cyclo ω8c	-	-	TR	TR
C_18:1_ ω9c	TR	TR	1.63	1.04
C_20:4_ ω6,9,12,15c	TR	-	-	-
Branched saturated				
iso-C_14:0_	1.28	2.39	TR	TR
iso-C_15:0_	3.79	8.80	7.89	3.75
iso-C_16:0_	7.53	8.42	2.77	3.57
iso-C_17:0_	2.05	3.24	2.01	2.84
anteiso-C_15:0_	45.71	50.30	61.41	45.42
anteiso-C_17:0_	7.52	5.21	7.92	12.46

TR, trace amount (<1.0%); -, not detected.

## Data Availability

The original contributions presented in the study are included in the article/Appendix A, further inquiries can be directed to the corresponding author.

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
