# Peer review of "Nitrogen-Fixing Paenibacillus haidiansis and Paenibacillus sanfengchensis: Two Novel Species from Plant Rhizospheres"

_microorganisms, 2024, doi:10.3390/microorganisms12122561_

Round 1

Reviewer 1 Report

Comments and Suggestions for Authors

In  an attachment letter

Author Response

The manuscript Paenibacillus haidiansis sp. nov. and Paenibacillus sanfengchensis sp. nov., two novel nitrogen-ffxing species isolated from the plant rhizosphere contributed by Weilong Zhang, Miao Gao, Rui Hu, Yimin Shang, Minzhi Liu, Peichun Lan, Shuo Jiao, Gehong Wei, Sanfeng Chen introduces two novel nitrogen-ffxing species, Paenibacillus haidiansis and Paenibacillus sanfengchensis, isolated from the rhizospheres of plants in China. Detailed genomic, phylogenetic, and phenotypic analyses support their classiffcation as new species within the genus Paenibacillus, with potential applications in agriculture. The proposed title of the manuscript is: Paenibacillus haidiansis sp. nov. and Paenibacillus sanfengchensis sp. nov., two novel nitrogen-ffxing species isolated from the plant rhizosphere.

First of all, it is important to highlight that this manuscript lacks Microorganisms or an MDPI template. This ffaw makes the analysis a little bit difficult. The manuscript has ffve tables, each one written differently from the others without any pattern.

Response: Thank you. The manuscript, including tables, was revised according to MDPI template.

This title is informative and reffects the content of the manuscript. However, it could be enhanced for clarity and conciseness. My suggestion could de: a) Discovery of Two Novel Nitrogen-Fixing Paenibacillus Species from Plant Rhizospheres or Nitrogen-Fixing Paenibacillus haidiansis and Paenibacillus sanfengchensis: Novel Species from Plant Rhizospheres.

Response: Thank you. Title was revised according to reviewer’s commnents.

The manuscript is generally well-written but contains a few areas where clarity or technical language can be improved. Example: "Genomic sequences of both strains" should replace "Genome sequences of both strains." The spelling of biological terms appears accurate. Speciffc gene names (e.g., nifH, nifK) and biological nomenclature are consistently correct. However, a more consistent use of Latin binomials in italics would enhance presentation.

Response: Thank you. It was revised.

The scientiffcally robust introduction presents a clear context for the study and outlines the relevance of nitrogen-ffxing Paenibacillus species in agricultural applications. However, the linkage between the introduction and the title could be improved by explicitly introducing Paenibacillus haidiansis and Paenibacillus sanfengchensis within the opening paragraphs. Furthermore, a more distinct framing of the study's novelty regarding these new species would benefft clarity.

Response: Thank you. It was revised.

Reviewing the Materials and Methods section, I found several missing citations for the manufacturer and model information for reagents and equipment. Speciffc examples: a) Genome sequencing platform: "Genomes of strains M1 and H32 were sequenced in Novogene Co. in China by using the Illumina PE150 platform." However, no part number or city details are provided. b) DNA extraction kit: The "TIANamp Bacteria DNA Kit" does not have the manufacturer’s city, country, or part number.

Response: Thank you. It was revised.

The results are generally well-written and thorough, presenting the ffndings. Tables and ffgures are appropriately used and easy to understand. One issue is that ffgure legends, such as for phylogenetic trees and chemical structures, could benefft from additional explanations to help readers interpret them without needing to refer back to the text. Then, the legend may have improved.

Response: Thank you. It was revised.

The discussion is too short and inadequately detailed. The authors could be further strengthened by making clearer comparisons between the novel species and closely related Paenibacillus species. The manuscript should also better address the broader implications of these ffndings for ffelds such as biotechnology or agriculture. This will enhance the paper's appeal to international journals of higher scientiffc standards.

Response: Thank you. It was revised.

Most references are recent, with over 70% published after 2015, demonstrating that the study engages with current literature.

Response: Thank you for your good evaluation.

Then, my opinion is Reconsider after major revisions. My main reason to this consideration are: a) Missing details in the Materials and Methods regarding reagent manufacturers, models, and locations must be addressed for rigor. b) The discussion could be improved by linking the novel species more strongly to broader scientiffc and agricultural implications.

Response: Thank you. It was revised. Especially, Discussion part is greatly improved.

Reviewer 2 Report

Comments and Suggestions for Authors

The manuscript describes the isolation and characterization of two bacterial strains proposed as novel species in the Paenibacillus genus. Bacterial strains were isolated in culture media without nitrogen. Subsequently, the nitrogen-fixing capabilities of both bacterial strains were tested through the acetylene reduction test. According to the phylogenetic analyses using the sequence of 16S rRNA, sequence similitudes were below 98% with their respective closest type species in the Paenibacillus genus, suggesting that isolated could be novel species. Genome sequence analysis of both bacterial strains lets us identify key genes implicated in the nitrogen-fixing process.

Authors´ need address following commentaries

Main commentaries:

Authors could use the template of Microorganisms Journal

The numbering of the lines could facilitate the manuscript review

More detail in the nitrogen-fixing test is needed

Authors could propose potential applications of these strains or the ecological role and benefits for the plants from which these strains were isolated, or future research

Tables 1 and 3 have very little data, they are not justified for the manuscript, the information can be described in the text of the manuscript

Additional commentaries:

In abstract the following fragments seems unnecessary “and below 97 % similarity with other recognized members at 16S rRNA gene sequences” and “and below 97 % similarity with other recognized members at 16S rRNA gene sequences”

Check the format in percentage, example, “97 %”, check and correct all in the manuscript

Explain in a better way where strains were deposited, if were

Most Keywords are included in the manuscript title

In the introduction, explain better “alter microbial population”

Add species name in “rice”

Correct format in “K2HPO4”

Correct format in “mg-1 protein h-1”

Review and homogenize redaction in “phylogenetic tree was constructed in MEGA7 using the Maximum Likelihood method [23].” and “hylogenetic tree was done by using the maximum likelihood (ML) using MEGA 7 software [23].”

Add a space in “algorithm[https”

Define “TLC” , “HPLC” and “PR” acronyms

The use of Paenibacillus polymyxa WLY78 (positive control) did not described in methods

Correct format “of 98.6.5%”

Correct format in Table 2, for better visualization

Author Response

The manuscript describes the isolation and characterization of two bacterial strains proposed as novel species in the Paenibacillus genus. Bacterial strains were isolated in culture media without nitrogen. Subsequently, the nitrogen-fixing capabilities of both bacterial strains were tested through the acetylene reduction test. According to the phylogenetic analyses using the sequence of 16S rRNA, sequence similitudes were below 98% with their respective closest type species in the Paenibacillus genus, suggesting that isolated could be novel species. Genome sequence analysis of both bacterial strains lets us identify key genes implicated in the nitrogen-fixing process.

Authors´ need address following commentaries

Main commentaries:

Authors could use the template of Microorganisms Journal

Response: Thank you. Yes, it was revised.

The numbering of the lines could facilitate the manuscript review

Response: Thank you. Yes, it was revised.

More detail in the nitrogen-fixing test is needed

Response: Thank you. In the manuscript, nitrogenase activity, nifH gene sequences (encoding a subunit of nitrogenase) and nif gene cluster determined by genome sequence analysis were provided. These data can fully demonstrate that te two bacteria are nitrogen-fixing bacteria.

Authors could propose potential applications of these strains or the ecological role and benefits for the plants from which these strains were isolated, or future research.

Response: Thank you. In Introduction and Discussion, potential applications of these strains were added.

Tables 1 and 3 have very little data, they are not justified for the manuscript, the information can be described in the text of the manuscript.

Response: Thank you. According to your comments, Tables 1 and 3 are deleted and the data within Table 1 and Table3 were in main text.

Additional commentaries:

In abstract the following fragments seems unnecessary “and below 97 % similarity with other recognized members at 16S rRNA gene sequences” and “and below 97 % similarity with other recognized members at 16S rRNA gene sequences” 

Response: Thank you. It was revised according to your comments.

Check the format in percentage, example, “97 %”, check and correct all in the manuscript

Response: Yes, It was revised.

Explain in a better way where strains were deposited, if were Most Keywords are included in the manuscript title    

Response: Thank you. It was revised.

In the introduction, explain better “alter microbial population”  

Response: Thank you. It was revised.

Add species name in “rice” 

Response: Thank you. It was revised.

Correct format in “K2HPO4”    

Response: Thank you. It was revised.

Correct format in “mg-1 protein h-1”

Response: Thank you. It was revised.

Review and homogenize redaction in “phylogenetic tree was constructed in MEGA7 using the Maximum Likelihood method [23].” and “hylogenetic tree was done by using the maximum likelihood (ML) using MEGA 7 software [23].”  

Response: Thank you. It was revised.

Add a space in “algorithm[https”

Response: Thank you. It was revised.

Define “TLC” , “HPLC” and “PR” acronyms

Response: Thank you. It was revised.

The use of Paenibacillus polymyxa WLY78 (positive control) did not described in methods

Response: Thank you. It was described in material and methods..

Correct format “of 98.6.5%”

Response: Thank you. It was revised.

 Correct format in Table 2, for better visualization

Response: Thank you. It was revised.

Reviewer 3 Report

Comments and Suggestions for Authors

The topic of the research presented in the manuscript "Paenibacillus haidiansis sp. nov. and Paenibacillus sanfengchensis sp. nov., two novel nitrogen-fixing species isolated from the plant rhizosphere" is important for the current context of expanding the isolated microbial resources with the ability of nitrogen fixing.

This trait of the two microbial species are important to identify new and more efficient nitrogen-fixers for the improvement of biological applications for nitrogen cycle.

There are some improvements that can be done to the current form of the manuscript.

Introduction - add at the end of this section a sentence or a paragraph, to explain the aim of the research, the reason and potential applications of this research.

Materials and Methods section offers all the necessary information to replicate the experiment.

The Results section explore all the findings of this research, providing all the results obtained from test applied to both strains, their reaction to biochemical test and even images of flagella.

Discussion section - this section is very short. It might be better to combine the results with discussions, and to add potential applications of the findings. Additional references can be added to connect this research with other international ones related to the two species isolated. 

Overall, the manuscript provides a good isolation presentation and deserve to be improved.

Author Response

Introduction - add at the end of this section a sentence or a paragraph, to explain the aim of the research, the reason and potential applications of this research.

Materials and Methods section offers all the necessary information to replicate the experiment.

The Results section explore all the findings of this research, providing all the results obtained from test applied to both strains, their reaction to biochemical test and even images of flagella.

Discussion section - this section is very short. It might be better to combine the results with discussions, and to add potential applications of the findings. Additional references can be added to connect this research with other international ones related to the two species isolated. 

Overall, the manuscript provides a good isolation presentation and deserve to be improved.

Response: Thank you. It was revised.

Round 2

Reviewer 1 Report

Comments and Suggestions for Authors

The manuscript was improved. However, figures 1 and 2 aren't in 300 DPIs, making it difficult to understand. In Figure 3, the legend and bar are of too low quality. Please replace this photo with a 300 DPI. The discussion is still too short for a manuscript published in a high-impact journal.

Author Response

The manuscript was improved. However, figures 1 and 2 aren't in 300 DPIs, making it difficult to understand. In Figure 3, the legend and bar are of too low quality. Please replace this photo with a 300 DPI. The discussion is still too short for a manuscript published in a high-impact journal.

Response: Thank you. Figures 1, 2 and 3 are revised as in 300 DPIs. Discussion part is revised. The added part in Discussion is marked in green blue color.

Reviewer 2 Report

Comments and Suggestions for Authors

After reviewing the new version of the manuscript, I consider that the authors have adequately addressed the comments from the first revision of the manuscript. There are no additional comments.

Author Response

After reviewing the new version of the manuscript, I consider that the authors have adequately addressed the comments from the first revision of the manuscript. There are no additional comments.

Response: Thank you.

Reviewer 3 Report

Comments and Suggestions for Authors

The authors have responded to all the comments and suggestions.

Author Response

The authors have responded to all the comments and suggestions.

Response: Thank you.